# Protein-Mediated Electroporation in a Cardiac Voltage-Sensing Domain Due to an nsPEF Stimulus

**DOI:** 10.3390/ijms241411397

**Published:** 2023-07-13

**Authors:** Alvaro R. Ruiz-Fernández, Leonardo Campos, Felipe Villanelo, Jose Antonio Garate, Tomas Perez-Acle

**Affiliations:** 1Computational Biology Lab, Fundación Ciencia & Vida, Santiago 7780272, Chile; lcamposp5@correo.uss.cl (L.C.); felipe@dlab.cl (F.V.); jgarate@dlab.cl (J.A.G.); 2Facultad de Ingeniería y Tecnología, Universidad San Sebastián, Santiago 8420524, Chile; 3Millennium Nucleus im NanoBioPhysics, Universidad de Valparaiso, Valparaiso 2351319, Chile; 4Centro Interdisciplinario de Neurociencia de Valparaíso, Universidad de Valparaíso, Valparaiso 2360102, Chile

**Keywords:** nsPEF, NPS, pores, complex pores, ionic channels, VSD, electroporation

## Abstract

This study takes a step in understanding the physiological implications of the nanosecond pulsed electric field (nsPEF) by integrating molecular dynamics simulations and machine learning techniques. nsPEF, a state-of-the-art technology, uses high-voltage electric field pulses with a nanosecond duration to modulate cellular activity. This investigation reveals a relatively new and underexplored phenomenon: protein-mediated electroporation. Our research focused on the voltage-sensing domain (VSD) of the NaV1.5 sodium cardiac channel in response to nsPEF stimulation. We scrutinized the VSD structures that form pores and thereby contribute to the physical chemistry that governs the defibrillation effect of nsPEF. To do so, we conducted a comprehensive analysis involving the clustering of 142 replicas simulated for 50 ns under nsPEF stimuli. We subsequently pinpointed the representative structures of each cluster and computed the free energy between them. We find that the selected VSD of NaV1.5 forms pores under nsPEF stimulation, but in a way that significant differs from the traditional VSD opening. This study not only extends our understanding of nsPEF and its interaction with protein channels but also adds a new effect to further study.

## 1. Introduction

nsPEF, a technology that emerged in 1995 [1], has seen a significant surge in research interest since 2005 [2]. nsPEF’s ability to elicit specific cellular effects [3] has resulted in an impressive range of applications, including the activation of neurons [4,5,6,7,8,9] and myocytes [10,11,12,13], wound healing [14,15,16,17], the manipulation of phenotype [18], the modulation of gene expression [19,20,21,22,23,24], antiparasitic effects [25,26,27], enhancement of the immune response [28,29,30,31,32,33], cell proliferation [18,34,35,36], improved fermentation [37,38], sterilization for the food industry [39,40,41], seed germination [42,43,44], and most notably, the development of novel cancer therapies [2]. Recently, nsPEF has been proposed for virus inactivation [45].

The nsPEF technique involves delivering high-electric-field pulses (∼1–0.3 V/nm) in nanoseconds or even picoseconds into biological tissues or cells, although the molecular mechanism is not entirely clear. Molecular dynamics (MD) studies have shown that nsPEF is capable of forming membrane nanopores [46,47], as was hypothesized several years before [48]. While direct evidence of nanopore formation is lacking due to their small size (~2 nm) and transient nature, indirect experimental results strongly support the formation of nanopores due to an nsPEF stimulus [5,49,50]. Further experimental results have shown that the activation of voltage-gated channels (VGCs) is another primary effect of nsPEF [4,8,51,52,53,54]. This observation is puzzling because nsPEF is much faster than the timescale on which the gating occurs in these channels (on the order of milliseconds) [55].

However, recent research using MD simulations has suggested an additional effect of nsPEFs: the creation of pores in transmembrane proteins. In 2018, the first study to investigate pore formation in transmembrane proteins (specifically, human aquaporin) using MD simulations was published [56]. Later, in 2020, Rems et al. [57] also used MD simulations to investigate nsPEF-induced pore formation in three distinct voltage-gated channels (a bacterial VGNC, a eukaryotic VGNC, and a human hyperpolarization-activated cyclic nucleotide-gated channel), observing both simple and complex pore formation in the VSD. Complex pores were proposed to be the source of theorized lipidic pores and may be stabilized by the presence of ions and other channel components, such as TMHs [58,59,60]. Recently, our group also observed the formation of complex pores in the VSD of a human VGCC using an MD simulation under the application of an external electric field of 0.2 V/nm lasting for 50 ns to mimic an nsPEF stimulus [61]. However, there is currently no direct experimental evidence for the formation of membrane nanopores or transmembrane pores due to nsPEF stimulation.

Ubiquitous and structurally similar integral membrane proteins, known as voltage-gated channels (VGCs), include Na^+^, Ca^2+^, and K^+^ channels [62]. While voltage-activated Na^+^ and Ca^2+^ channels consist of monomers with varying auxiliary subunits, voltage-activated K^+^ channels are tetramers. All of these channels have four repeated structures, each formed by six 
α
-helices, as shown in Figure 1. The VSD is composed of helices S1 to S4, while the pore domain is formed by helices 5 and 6 [63]. Notably, the four VSD structures are present in all Na^+^, Ca^2+^, and K^+^ channels [64].

Recently, nsPEF stimulation was proposed as a new defibrillation method to achieve a higher efficiency of defibrillation in the first shock [65]. Among several effects (reduced shock energy, minimized side effects, lowered probability of the reinduction of arrhythmias), the transient inhibition of Na^+^ and Ca^2+^ VGCs [66,67] may aid in the antiarrhythmic effect of nsPEF defibrillation. Therefore, it is relevant to study, at a molecular level, the effects of nsPEF on a VGC specific for cardiac tissue.

In this article, we investigate the effects of a mimic of nsPEF on the VSD of the Nav1.5-E1784K voltage-gated sodium channel (VGNC) using MD simulations. Nav1.5 is the primary VGNC channel in the heart, and mutations in Nav1.5 have been linked to various cardiac disorders, including type 3 long QT syndrome (LQT3) and Brugada syndrome (BrS). The E1784K mutation is commonly found in patients with both LQT3 and BrS [68]. The Nav1.5 VGNC used in our study is in its native conformation and is not bound to any non-protein molecules. This mutated VGNC has a structure nearly identical to that of the wild-type human Nav1.5. Interestingly, the VSD of Nav1.5 selected has three arginines in its S4 helix instead of four, as is typical in most VGCs. These arginines play a crucial role in the activation of VGCs [69,70,71,72,73,74], as they provide a high charge density that can respond to changes in membrane potential.

## 2. Results

### 2.1. Exploring Electric Field Magnitudes

Experiments involving nsPEF on tissues or cells encompass variations in pulse intensity, duration, time interval between pulses, and the number of pulses. For example, to activate neurons, an nsPEF protocol of 27.8 kV/cm with a single pulse duration of 10 ns is utilized [6]. Conversely, to enhance germination, 20 nsPEF pulses with an intensity ranging from 10 to 30 kV/cm with a pulse duration of 100 ns [43] were applied. Additional protocols related to other applications of nsPEF have been reviewed previously, with extensive details on the experimental setups used in each case [3].

We performed an exploratory analysis using MD by applying different 
$\vec{E_{z}}$
 values along the z-axis to a box containing an equilibrated cardiac VSD embedded in a POPC bilayer. Our goal was to determine the optimal 
$\vec{E_{z}}$
 that would induce the most significant structural changes in the VSD, leading to the formation of both protein pores and complex pores, without perturbing the bilayer arrangement of the membrane or denaturing the protein. The 
$\vec{E}$
 values used ranged from 0.1 V/nm to 0.2 V/nm in increments of 0.02 V/mm. To understand the resulting potential across a heterogeneous system like this, please refer to [75].

The chosen range of the electric field magnitude was not arbitrary. Previous studies have employed voltage-sensitive dyes [76,77] and microelectrodes [78] to determine that a membrane potential of 1.5 V represents the largest potential that the membrane can support before discharging. Other studies support this voltage value [76,77,79,80,81,82,83]. We therefore explored 
$\vec{E}$
 values up to 0.2 V/nm, given that the membrane potential for an 
$L_{z}$
 value of 8.5 nm is 1.7 V, which is near the theoretical limit of 1.5 V. We quantified the structural changes in the VSD induced by each 
$\vec{E}$
 value by measuring the RMSD of eight replicas per 
$\vec{E}$
 value.

At an electric field strength of 
$\vec{E_{z}}$
 = 0.10 V/nm and 
$\vec{E_{z}}$
 = 0.12 V/mn, the VSD exhibits negligible changes in RMSD across each of the replicas (Figure 2A,B). This suggests that the internal forces governing the structural integrity of the VSD remain resilient to external stimuli up to a certain threshold. Conversely, when exposed to electric fields of 
$\vec{E_{z}}$
 = 0.14 V/nm or greater, the majority of the replicas displayed RMSD values higher than 3 nm, which is probably the result of the denaturation of the VSD. In addition, the bilayer shows a 90° rotation in the x-y plane, yielding a model that has no biological significance (Figure 2C–F). In Appendix A, we illustrate the bilayer’s turning point in response to external electric fields ranging from 
$\vec{E_{z}}$
 = 0.14 up to 0.20 V/nm. For this reason, we selected a magnitude of 
$\vec{E_{z}}$
 = 0.13 V/nm as optimal to observe changes in the VSD structure without unfolding the bilayer or the protein. To exhaustively sample the phase space, we ran 200 new replicas with the selected electric field of 
$\vec{E_{z}}$
 = 0.13 V/nm. From these replicas, the highest RMSD at the final point of the simulation was 2.57 nm and exhibited a POPC bilayer that retained its original arrangement, although some of them encompass important membrane rearrangements, as can be observed in Figure 3. To observe an example of pore formation as a function of RMSD extracted from the 200 replicas, please refer to Appendix A. To observe examples of VSDs structures with water crossing the membrane at different final RMSDs see Figure 4.

### 2.2. Clustering VSD

To identify the most representative conformational changes in the voltage-sensing domain (VSD), we employed a clustering approach based on an unsupervised machine learning algorithm. This analysis involved the calculation of the fraction of native contacts (FNC) and the RMSD between different structures of the VSD extracted from 200 new replicas at 
$\vec{E_{z}}$
 = 0.13 V/nm.

A total of 142,000 frames were obtained from the 142 replicas, given that the total time of the simulation was 10 ns, and frames were extracted every 10 ps. This yielded a total of 142,000^3^ = 2.86 × 10^15^ parameters or coordinates, which was computationally unmanageable. To address this issue, we reduced the number of frames by obtaining frames every nanosecond, resulting in 10 frames every 10 ns. The RMSD of the 142 replicas can be observed in Appendix A.

After the clustering method, we obtained three clusters. Cluster 1 contains 2,269,895 tuples, cluster 2 contains 2,200,065 tuples, and cluster 3 contains 1,146,558 tuples (Figure 5). A tuple is a data structure constructed from the three parameters described in the Section 4 (Equation (Equation 4)). Our clustering analysis indicates that the data are well represented by these three groups, with two clusters being very similar to each other, both presenting simple pores. The VSDs of cluster 2 present slightly more open VSDs than the VSDs of cluster 1, allowing more water to cross. The VSDs of cluster 3 present more open pores that form complex pores. Regarding the density map in Figure 6, cluster 3 appears to be the largest, but this is actually due to the data being more dispersed, as can be observed in Figure 5D.

From the density profile of each cluster, the closest tuple to the maximum density was obtained using the least-squares method. This method was used with a kernel density estimation (KDE) constructed with all tuples using the quadratic error to find the tuple closest to the maximum density point in the KDE. Each of these tuples has an associated frame from the simulations that corresponds to the most representative conformation for each cluster. Figure 7 displays VSDs representative of each cluster.

Considering the number of executed independent replicas and produced frames (over 5 million points), we think that the phase space describing the conformational changes in the VSD under the influence of external stimuli mimicking nsPEF has been thoroughly sampled. The total set of frames collectively represents the kinetically accessible set of possible conformations of the molecular system. Therefore, it can be assumed that all possible conformations have been sampled, resulting in an equilibrium between clusters. The free energy between clusters can now be computed by relying on established methods. To see how the tuples cover the phase space, refer to Appendix A.

(1)
$$\Delta G^{0}=-RTln\left(\frac{P1}{P2}\right),$$

where *R* is the molar gas constant, *T* is the temperature, and in our case, 
P1
 and 
P2
 represent the number of tuples in each cluster. The calculated 
Δ
G values between all clusters appear in Table 1.

These values are subject to thermal noise, and thus, the VSD representatives of each cluster are in continuous transition. A 2D representation depicting the density of the tuples for each cluster, considering the RMSD and the FNC, can be seen in Figure 6.

## 3. Discussion

The classical activation of voltage-gated ionic channels occurs due to the movement or torsion of the S4 helix of the VSD, which is triggered by a change in the membrane potential. This phenomenon is attributed to the highly charged nature of the S4 helix, which contains four charged arginine residues in its structure. Various models have been proposed to explain the transfer of charge during the activation of ionic channels, and all of them are linked to the movement of the S4 helix [84,85]. However, our results indicate that the formation of simple and complex pores in response to an nsPEF stimulus does not depend on the S4 helix. A possible explanation is that the selected VSD, Nav1.5, contains only three arginine residues, in contrast to most VSDs, which have four arginine residues. This change in the net positive charge of the S4 helix can make the protein behave differently under nsPEF stimuli.

Another explanation is that the formation of pores under nsPEF is a phenomenon completely different from the classical VGC activation via membrane potential. This explanation is supported by the fact that the classical activation of VGCs occurs at the scale of microseconds, which is three orders of magnitude slower than the nsPEF pulse duration (nanoseconds).

Regardless of the explanation, our results indicate that the selected VSD, Nav1.5-E1784K, forms pores with less energy than was found in other studies with similar systems. Rems et al. found that three different VGCs in the POPC bilayer required 1.5 V to form complex pores [57]. In a previous study by our group, an electrical potential of 1.75 V was necessary to form complex pores in the VSD from a calcium VGC in a more resistant membrane with cholesterol (3:1 POPC–cholesterol ratio) [61]. In the present study, an electric potential of 1.1 V (0.13 V/nm × 8.5 nm) was necessary, indicating that this specific VSD is more prone to pore formation. While it is possible that a lower number of arginine residues in the S4 helix may increase the susceptibility to pore or complex pore formation, more studies are required to confirm this conclusion.

One important observation is that the replicas in which the VSD underwent bigger structural changes were accompanied by the rearrangement of the surrounding membrane. Some of these rearrangements are important (Figure 3). Nonetheless, this occurs in a few cases (Figure 7). Previous studies have shown that membranes can act as allosteric regulators of protein structure and function [86,87], and thus, we can hypothesize that the formation of pores and complex pores in VSDs upon nsPEF involves the cooperative rearrangement between the protein and the membrane. This is supported by our previous study that showed pores and complex pores forming in a cholesterol–POPC 3:1 membrane at an electric field strength of 
$\vec{E_{z}}$
 = 0.2 V/nm, but not in a pure POPC membrane [61].

Of note, further research must be conducted to account for the involvement of the bilayer on nsPEF-induced pore formation, which includes the size of the bilayer and the incorporation of a complex mix of lipids, more similar to biological membranes.

Finally, there is a possibility that under nsPEF stimuli, VSD undergoes structural changes that lead to complex protein–lipid pores, which work differently from the classical opening–closing mechanism described for these type of channels previously. Understanding the formation of this type of pore in cardiomyocytes due to an nsPEF stimulus may contribute to comprehending the biophysics of using nsPEF as a defibrillator, which is a promising innovation for emergency cardiac therapy.

## 4. Materials and Methods

### 4.1. Model Preparation and MD Simulations

The simulations were conducted using the GROMACS molecular dynamics package 2020.6 [88], and the CHARMM36 force field was utilized to account for parameters, including bond length, angle bending, angle torsion, and non-bonded interactions of the molecular system [89]. The selected VSD, Nav1.5-E1784K, was embedded in a POPC bilayer containing 166 POPC molecules using the Membrane Builder module of the online software CHARMM-GUI [90]. The resulting system was subsequently introduced into a rectangular simulation box and filled with 9713 H_2_O molecules and 1 K^+^ ion to neutralize the system. The VSD used in the study (residues 312-435) was obtained from electron microscopy of a human Nav1.5-E1784K channel (PDB ID: 7DTC), with a resolution of 3.30 Å [68].

Prior to the molecular dynamics simulation, the steepest descent minimization algorithm was applied to relax the molecular system, followed by 6 steps of equilibration. The dimensions of the simulation box were 7.79 nm × 7.79 nm × 8.50 nm, which ensured that the VSD did not see its periodic images. See Appendix A.

The leap-frog integrator algorithm was utilized with a time step of 2 fs. The Particle Mesh Ewald (PME) method was used to calculate long-range electrostatic interactions [91]. The interactions were truncated using a cut-off of 1.2 nm for non-covalent forces. The bond lengths were constrained using the linear constraint solver (LINCS) [92]. The simulations were performed in an NPT ensemble at a temperature of 310 K and a pressure of 1 bar, coupled to a Nose–Hoover thermostat [93] and a Parrinello–Rahman barostat [94], respectively. Time constants of 1 ps and 5 ps were used for temperature and pressure, respectively. Periodic boundary conditions were applied in all directions. The TIP3P water model [95] was used for all simulations. For all simulations performed with replicas, the initial velocities were randomized from a Maxwell distribution.

The system resulting from equilibration was further simulated for 200 ns in an NPT ensemble (Figure 8). The resulting structure was subsequently simulated, but now with the addition of an external electric field along the z-axis (perpendicular to the bilayer) of different magnitudes (
$\vec{E_{Z}}$
) to determine the optimal one capable of creating simple and complex pores in the VSD.

During the simulation, 
$\vec{E}$
 was implemented as a constant positive force that was applied to all charged atoms (integral or fractional charge) in the system [75]. The resulting force (
$F_{ia}$
) is defined as follows:
(2)
$${\vec{F}}_{ia}=q_{ia}\;{\vec{E}}_{z}$$

where 
$q_{ia}$
 is the partial atomic charge of atom *i*. The total potential 
$\Delta V_{Z}$
 across the simulation box exerted by the electric field follows the relation:
(3)
$$\Delta V_{Z}={\vec{E}}_{z}\;L_{z}$$

where 
$L_{z}$
 is the length of the box along the *z*-axis.

Since nsPEF is a quadratic pulse, only the strength of the electric field is required.

The RMSD was calculated between all atoms of the VSD. The FNC was calculated using MDAnalisis, using a hard cutoff of 8 Å for determining native contacts [96].

### 4.2. Clusterization

The clustering of VSD structures was based on pairwise comparisons of frames extracted from all replicas. Briefly, a three-dimensional space was constructed using three parameters extracted from frame-to-frame pairwise comparisons. This tridimensional parameter is defined as:
(4)
$$P_{\left(x,y\right)}=\left[RMSD_{\left(x,y\right)},FNC_{\left(x,x_{0}\right)},FNC_{\left(y,y_{0}\right)}\right]$$

where 
$P_{\left(x,y\right)}$
 is the tridimensional parameter for the pair of frames *x* and *y*, RMSD is the root mean square deviation, and FNC is the fraction of native contacts of each frame with respect to 
$x_{0}$
 and 
$y_{0}$
, which correspond to the first frame in the respective replica.

A filtering step was included to discard simulations that did not attain a stable structure. To do so, we assessed the change in the first derivative of the RMSD over the last 10 ns of the simulation, and only those simulations with a change of less than 15% were kept. We selected 162 stable simulations that met this criterion. We further filtered these simulations to exclude those that ended with an RMSD of less than 0.35 nm and did not form a conclusive VSD pore as depicted in Figure 9. This filtering step yielded a final set of 142 simulations that were suitable for clusterization from the initial 200 simulations (replicas) available. This filtering step improved the quality of the clusterization by ensuring that the resulting structures had full VSD pores.

Then, we normalized the data using a scaling method from the scikit-learn module in Python [97]. The weight of each tuple was different, since tuples could share the same coordinates. Finally, clustering was performed using the k-means algorithm [98], with the elbow method [99] used to determine the optimal number of clusters, which was three. To calculate density profiles for clusters, the KDE method was used from the scikit-learn module in Python [97].

## 5. Conclusions

The primary determinant governing the formation of pores in the VSD due to nsPEF stimulation is not the S4 helix, as one might expect intuitively. This specific cardiac VSD, containing only three arginines in the S4 helix, is less resistant to external electric fields compared to those of similar studies [57,61]. 
$\vec{E_{z}}$
 values below 0.14 V/nm are able to form complex protein–lipid pores without denaturing the bilayer or protein. Full VSD pores start forming at an RMSD greater than 0.35 nm.

We propose a clusterization procedure for VSD structures that yields three clusters, and the most representative VSDs of each cluster are not very different from one another. However, the way in which water passes through them is very different: VSD representatives from two clusters form simple pores, while the VSD representative of cluster 3 presents a complex pore. Thermal noise of 2.47 kJ/mol indicates that all representative structures of the VSD obtained through clusterization oscillate in thermal equilibrium with each other.

The observation of protein-mediated electroporation through the formation of pores and complex pores in VSDs in response to nsPEF stimulation, as demonstrated in this MD simulation study, must be considered for a better understanding of the biophysics underlying the effects of nsPEF applications, especially in the case of cardiac VSDs. This phenomenon of protein-mediated electroporation due to an nsPEF stimulus is not well described and should be considered as a possible effect of nsPEF in cells. We hope that further experimentation can confirm or dismiss this finding.

## Figures and Tables

**Figure 1 ijms-24-11397-f001:**
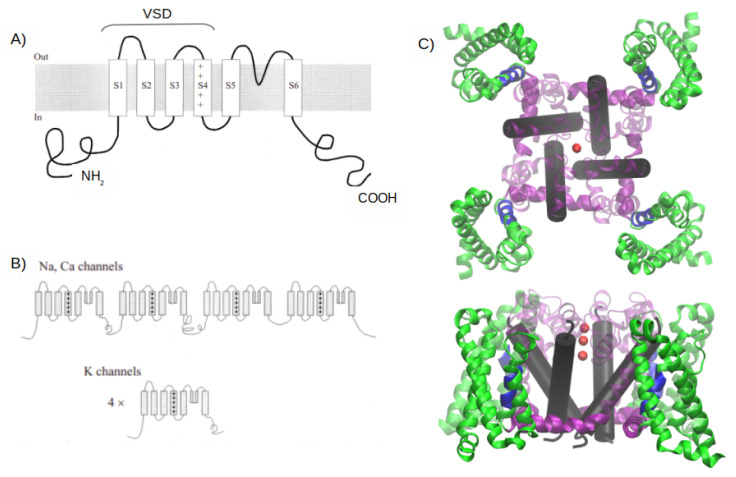
Structural overview of voltage-gated channels. (**A**) Transmembrane disposition of a single voltage-gated K^+^ channel subunit. There are six transmembrane regions, denoted by S1 to S6. (**B**) Structural organization of voltage-gated channels. Na^+^ and Ca^2+^ channels have four homologous repeats of the core motif in a single polypeptide chain; K^+^ channels are tetrameric assemblies of subunits with a single core motif. (**C**) A 3D representation of the crystallographic structure of a *Arcobacter butzleri* RM4018 VGCC in its open state (PDB ID: 4MS2). In green are the four VSDs, in blue is the helical conformation 3_10_ in the S4 helix, in purple are the S4-S5 helices (not present in (**A**) or (**B**)) that connect the S5 and S6 helices, and in light purple are the S5 helix and the rest of the amino acids that connect to the S6 helix. In the black cylinder is the S6 helix, and the red spheres represent three calcium atoms. Figure extracted from [61].

**Figure 2 ijms-24-11397-f002:**
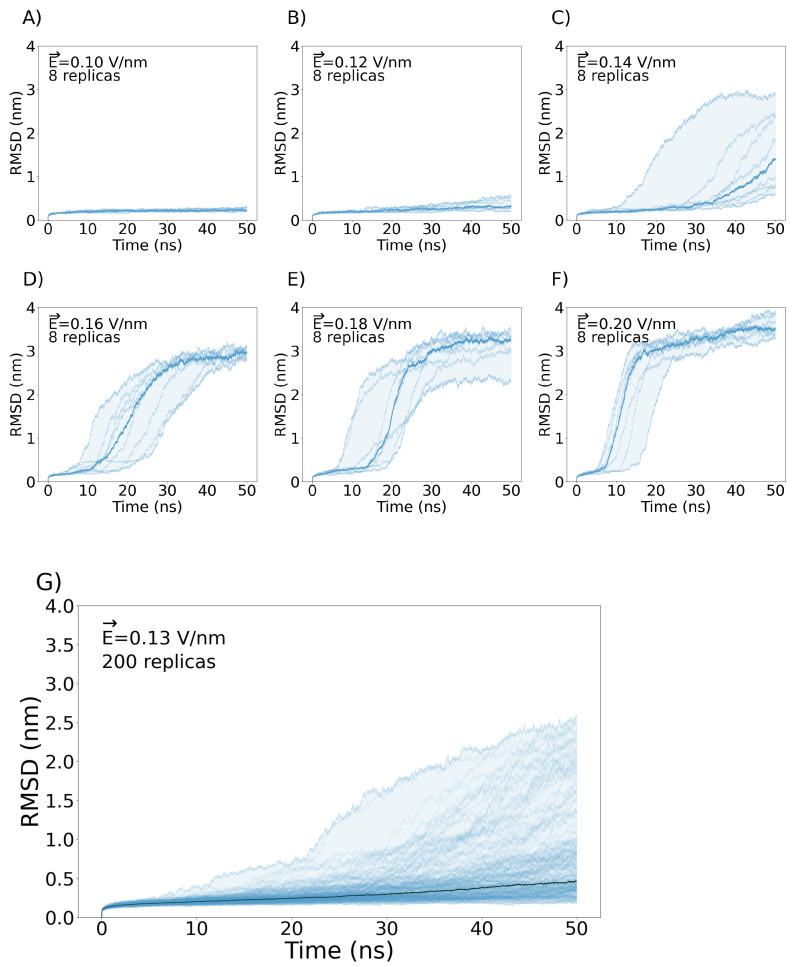
Evolution of Nav1.5-E1784K VSD RMSD during the simulations using different 
$\vec{E_{z}}$
 values. (**A**) Eight replicas at 
$\vec{E_{z}}$
 = 0.10 V/nm. (**B**) Eight replicas at 
$\vec{E_{z}}$
 = 0.12 V/nm. (**C**) Eight replicas at 
$\vec{E_{z}}$
 = 0.14 V/nm. (**D**) Eight replicas at 
$\vec{E_{z}}$
 = 0.16 V/nm. (**E**) Eight replicas at 
$\vec{E_{z}}$
 = 0.18 V/nm. (**F**) Eight replicas at 
$\vec{E_{z}}$
 = 0.20 V/nm. (**G**) Two hundred replicas at 
$\vec{E_{z}}$
 = 0.13 V/nm. The bold line in each plot represents the mean of each distribution of RMSD as a function of time.

**Figure 3 ijms-24-11397-f003:**
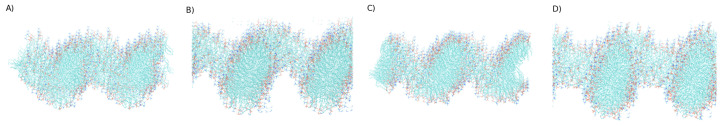
Membrane rearrangement of replicas with higher final RMSDs forming complex pores under an external 
$\vec{E_{z}}$
 = 0.13. Views are from the bilayer plane. Protein and water are not shown. (**A**) Final RMSD = 2.38 nm. (**B**) Final RMSD = 2.25 nm. (**C**) Final RMSD = 2.47 nm. (**D**) Final RMSD = 2.22.

**Figure 4 ijms-24-11397-f004:**
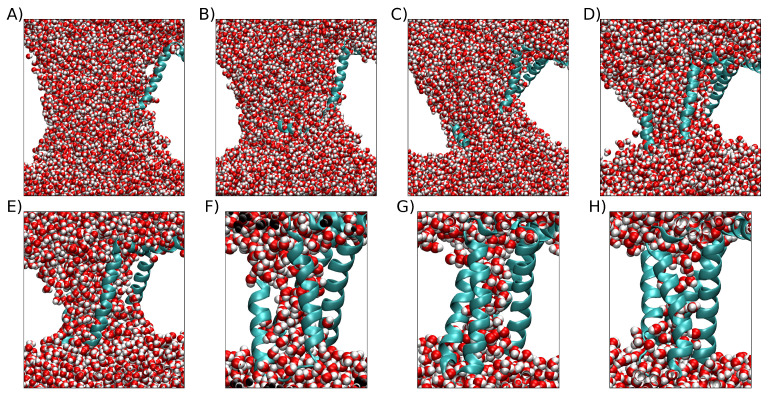
Examples of VSDs structures with water crossing the membrane at different final RMSDs. The protein is represented by the sky-blue cartoon, and water is represented by van der Waals spheres. The white space is occupied by the POPC bilayer, which is not shown for clarity. (**A**) Example of VSD with large RMSD. (**B**) Example with higher RMSD below 2.0 nm. (**C**) Example with higher RMSD below 1.6 nm. (**D**) Example with higher RMSD below 1.2 nm. (**E**) Example with higher RMSD below 0.8 nm. (**F**) Example with higher RMSD below 0.6 nm. (**G**) Example with higher RMSD below 0.4 nm. (**H**) Example with higher RMSD below 0.4 nm.

**Figure 5 ijms-24-11397-f005:**
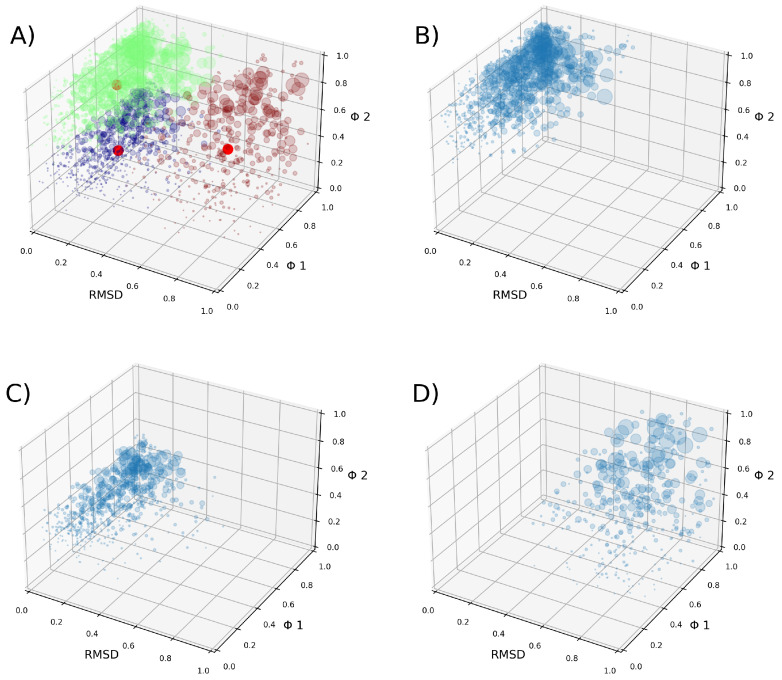
(**A**) Obtained clusters of structures are shown in 3D plots. Each circle represents one tuple, with its size being proportional to its weight. The different clusters are shown in different colors, and their centroids are represented by red circles. In green is cluster 1, in purple is cluster 2, and in brown is cluster 3. (**B**–**D**) Each cluster is plotted on its own.

**Figure 6 ijms-24-11397-f006:**
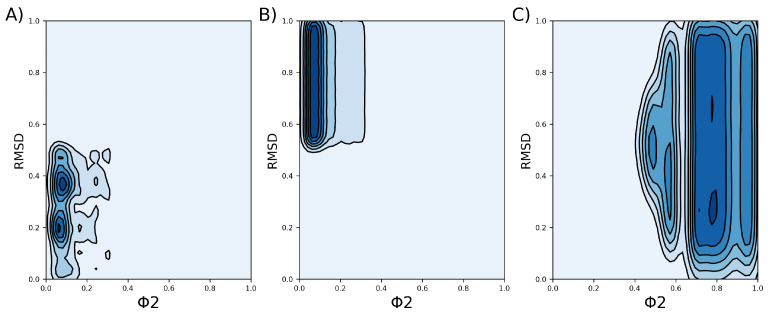
Density profile of each cluster, which was obtained based on the RMSD and the FNC parameter, denoted by 
ϕ
. Darker colors indicate regions of higher density. (**A**) Density profile of cluster 1. (**B**) Density profile of cluster 2. (**C**) Density profile of cluster 3.

**Figure 7 ijms-24-11397-f007:**
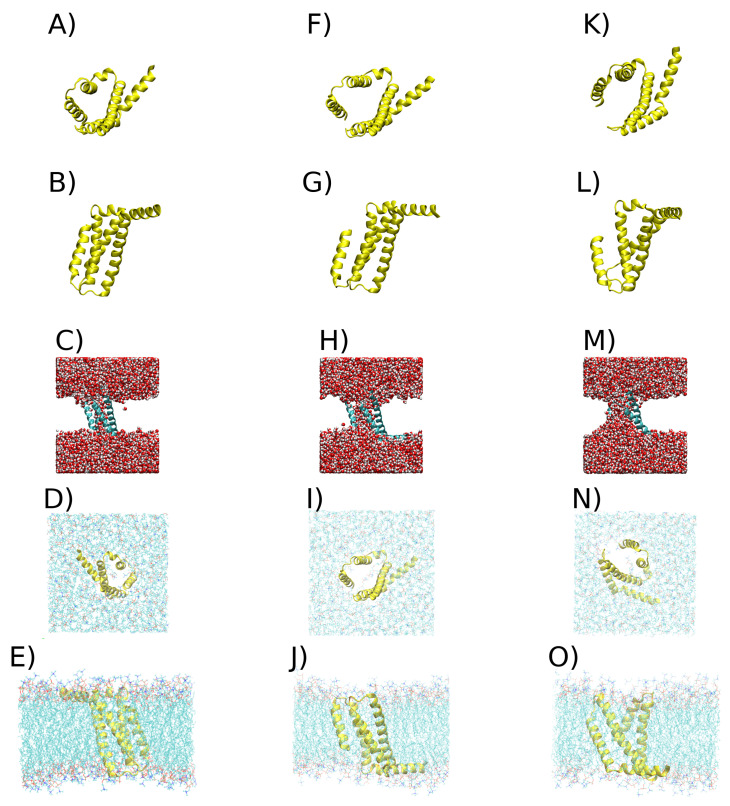
Representative structures of VSDs in each cluster. (**A**) VSD of cluster 1 in yellow cartoon, viewed from above the x-y plane. (**B**) VSD of cluster 1 in new cartoon in yellow, viewed from above the z-y plane. (**C**) VSD of cluster 1 in new cartoon, with water represented by van der Waals spheres. (**D**) VSD of cluster 1 in new cartoon, viewed from above the x-y plane, with the membrane represented by lines. (**E**) VSD of cluster 1 in new cartoon, viewed from above the z-y plane, with the membrane represented by lines. (**F**) VSD of cluster 2 in yellow cartoon, viewed from above the x-y plane. (**G**) VSD of cluster 2 in yellow cartoon, viewed from above the z-y plane. (**H**) VSD of cluster 2 in new cartoon, with water represented by van der Waals spheres. (**I**) VSD of cluster 2 in new cartoon, viewed from above the x-y plane, with the membrane represented by lines. (**J**) VSD of cluster 2 in new cartoon, viewed from above the z-y plane, with the membrane represented by lines. (**K**) VSD of cluster 3 in yellow cartoon, viewed from above the x-y plane. (**L**) VSD of cluster 3 in yellow cartoon, viewed from above the z-y plane. (**M**) VSD of cluster 3 in new cartoon, with water represented by van der Waals spheres. (**N**) VSD of cluster 3 in new cartoon, viewed from above the x-y plane, with the membrane represented by lines. (**O**) VSD of cluster 3 in new cartoon, viewed from above the z-y plane, with the membrane represented by lines.

**Figure 8 ijms-24-11397-f008:**
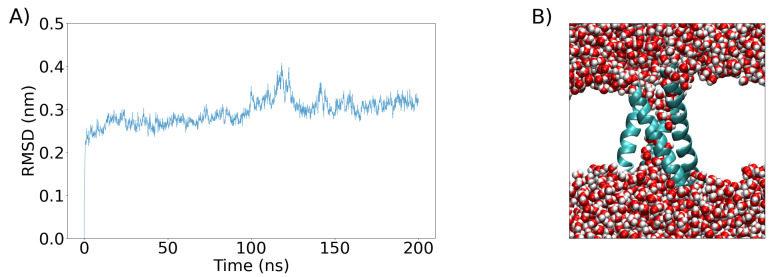
(**A**) The RMSD of the VSD as a function of time calculated during a 200 ns equilibration period (see Section 4 for details). (**B**) The VSD conformation after the 200 ns equilibration period is shown as a cartoon representation and colored sky blue, while water molecules are represented as van der Waals spheres. The white space is occupied by the POPC bilayer, which is not shown for clarity.

**Figure 9 ijms-24-11397-f009:**
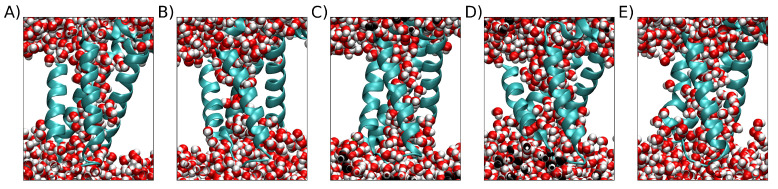
Five examples of replicas with higher final RMSDs under 0.35 nm. The protein is represented by the sky-blue cartoon, and water is represented by van der Waals spheres. The white space is occupied by the POPC bilayer, which is not shown for clarity. (**A**) Final RMSD = 0.349 nm. (**B**) RMSD = 0.323 nm. (**C**) Final RMSD = 0.330. (**D**) Final RMSD = 0.328 nm. (**E**) Final RMSD = 0.348 nm.

**Table 1 ijms-24-11397-t001:** Free energy values between clusters.

Clusters	$\Delta G^{0}$ (J/mol)
1–2	−80.5
1–3	−1760.2
2–3	−1679.7

## Data Availability

All scripts and data can be found at https://github.com/DLab/article_nspef_pores (accessed on 7 July 2023).

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
