# Peer review of "Protein-Mediated Electroporation in a Cardiac Voltage-Sensing Domain Due to an nsPEF Stimulus"

_ijms, 2023, doi:10.3390/ijms241411397_

Round 1
Reviewer 1 Report
See the pdf file uploaded.

See the pdf file uploaded.
Reviewer 2 Report
Ruiz-Fernandez et al proposed a “new” (in silico) phenomenon of protein-mediated electroporation in a Nav VSD by repeating molecular dynamic simulations of isolated VSDs in virtual membranes during short-stimulation of nsFEF and clustering obtained conformations of the VSD. The calculations were carried out apparently in nonphysiological conditions (> 0.1 V/nm) that cannot be tolerated by normal biological membranes. The proposed dilation of the 4-helix bundle of the VSD under such conditions is energetically unlikely and appears not even marginally relevant to any known measurements of VSDs in biomembranes. The forcefield in the calculations is not well calibrated under the exceedingly strong electric field used here either. The whole study and the results are therefore of littler or no biological relevance to the conformational transitions of the VSDs from well-known voltage-gated ion channels. The results are premature and must be experimentally validated to avoid misleading the field or the audience interested in novel properties of VSDs.
Some of the main concerns are listed in the next.
1. Electroporation in bilayer membranes is not new. It is important to separate pore formation in lipid membranes from that in protein in the system. The MD system appears to have too small lipid bilayers for such a purpose.
2. The electrical conditions used in the calculations lead to severe distortions in bilayers (Fig. 4). Is it possible that the VSDs need to rearrange themselves in adaptation to such changes and thus the protein-mediated pores would not happen when the lipid bilayers are larger in surface area and are distorted.
3. The forcefield used in calculations appears to be suitable for near-physiological conditions. Is it suitable for the atoms in the very high electric field (~0.13 V/nm, equivalent to ~0.5 V/bilayer)? Do water and lipid molecules behave the same as in a low field of less than 0.1 V/membrane? Are polarization and bond-breakage becoming an issue under such conditions?
4. Lipids were not explicitly included in the models in Fig. 2, 5, 6. Are they behaving differently, such as jumping out of the bilayer boundary or is the bilayer well maintained?
5. It is well known that annular lipids around a voltage-gated ion channel affect the energetics of the VSDs significantly and thus give rise to the so-called “lipid-dependent gating” of these channels (Zheng, H. Nat. Comm. 2011). Can such a lipid effect on the VSD be calculated properly? Can the lipid bilayer suppress the dilation of the helical bundle in a VSD?
6. The energetics involved in the calculations in 1-3 and 2-3 (Table 1) are much larger than ~6-10 kcal/mol for physiological transitions of a VSD between its down and up conformations. It is important to test the relevance of these transitions to the physiology of the VSD.
7. The complex pores and even the enclosed pores in Fig.6, and 9 appear to result from unfolding of the VSDs. How likely is unfolding of VSD used in physiology?
8. Clustering showed in Fig. 7 seems show much larger RMSD within cluster 3 than the RMSD difference between clusters 1 and 2. Is there any intrinsic bias in classification?
9. there are multiple language issues. Such as: lines 253 – 255, lines 168-171, 186-188, etc.
Multiple places have sentences not properly structured.
Reviewer 3 Report
Minor revision needed
The first letter of the keywords should be capital and separated by semicolon.
Please recheck the affiliations in supplementary file.
Experimental part should be provided.
Reviewer 4 Report
See attachment for my feedback.

Here-and-there small grammar/typographical errors.
Round 2
Reviewer 1 Report
See the uploaded pdf file.

Reviewer 2 Report
The revision did not address my concerns well even though some changes were introduced. Besides, there are too many grammatical errors in the authors' responses, which affect the precision of their meanings. For example, their first sentence, "Us says in the article in lines..." was not properly constructed.
The argument that the membrane could tolerate 1.5 V in literature is different from the 0.13 V/nm in the nanometer scale, because a voltage applied macroscopically is not the same as the voltage falloff across a membrane. With patch-clamp and bilayer membrane recordings, it is well-known that typical biological membranes can not tolerate 0.1 V/nm. Electroporation of cell membranes is known to happen in such voltages.
The size of the box in the MD system is arbitrary, and the lipids around individual VSDs can be adjusted to include sufficient lipid bilayer areas around. What is said about ".. the VSDs don't see each other .." or " .. the minimum distance between VSDs periodic images ..." makes no good sense for not changing the system in Fig. 2B.
Because the authors agreed that larger membranes may generate different results, such membranes should be considered because they are more comparable to channels in cell membranes and to the observed electroporation or electrical breakdown. It is thus important to know if the membrane breakdown happens before VSD dilation (pore) does. Data in Figure 4 showed severe membrane distortions that would happen when larger lipid bilayers are included around each VSD.
If nsPEF could fragment DNA, it should be tested whether protein bonds could be broken by such strong fields. Because damage of peptide bonds could seriously hurt the study, it is not rigorous by adding the content in lines 388-91. It is thus important to validate the force field and have all lipid molecules modeled properly in the system (more than what is showed in Fig. 2B).
Too many errors in the rebuttal made it difficult to read and understand.
Round 3
Reviewer 1 Report
The model used by the authors is not self-consistent.
To some extent, this is inevitable, because the system is
complex. In the revised manuscript, the corresponding
shortcomings of the work are not hidden, and it will not
mislead readers. With these reservations, I believe that
the manuscript can now be accepted for publication.
None.